# GEFL: Extended Filtration Learning for Graph Classification

**Simon Zhang**
Purdue University
zhan4125@purdue.edu

**Soham Mukherjee**
Purdue University
mukher26@purdue.edu

**Tamal K. Dey**
Purdue University
tamaldey@purdue.edu

## Abstract

Extended persistence is a technique from topological data analysis to obtain global multiscale topological information from a graph. This includes information about connected components and cycles that are captured by the so-called persistence barcodes. We introduce extended persistence into a supervised learning framework for graph classification. Global topological information, in the form of a barcode with four different types of bars and their explicit cycle representatives, is combined into the model by the readout function which is computed by extended persistence. The entire model is end-to-end differentiable. We use a link-cut tree data structure and parallelism to lower the complexity of computing extended persistence, obtaining a speedup of more than 60x over the state-of-the-art for extended persistence computation. This makes extended persistence feasible for machine learning. We show that, under certain conditions, extended persistence surpasses both the WL[1] graph isomorphism test and 0-dimensional barcodes in terms of expressivity because it adds more global (topological) information. In particular, arbitrarily long cycles can be represented, which is difficult for finite receptive field message passing graph neural networks. Furthermore, we show the effectiveness of our method on real world datasets compared to many existing recent graph representation learning methods.[1]

## 1 Introduction

Graph classification is an important task in machine learning. Applications range from classifying social networks to chemical compounds. These applications require global as well as local topological information of a graph to achieve high performance. Message passing graph neural networks (GNNs) are an effective and popular method to achieve this task.

These existing methods crucially lack quantifiable information about the relative prominence of cycles and connected component to make predictions. Extended persistence is an unsupervised technique from topological data analysis that provides this information through a generalization of hierarchical clustering on graphs. It obtains both 1- and 0-dimensional multiscale global homological information.

Existing end-to-end filtration learning methods [1, 2] that use persistent homology do not compute extended persistence because of its high computational cost at scale. A general matrix reduction approach [3] has time complexity of $O((n + m)^\omega)$ for graphs with $n$ nodes and $m$ edges where $\omega$ is the exponent for matrix multiplication. We address this by improving upon the work of [4] and introducing a link-cut tree data structure and a parallelism for computation. This allows for $O(\log n)$ update and query operations on a spanning forest with $n$ nodes.

We consider the expressiveness of our model in terms of extended persistence barcodes and the cycle representatives. We characterize the barcodes in terms of size, what they measure, and their expressivity in comparison to WL[1] [2]. We show that it is possible to find a filtration where one of its cycle's length can be measured as well as a filtration where the size of each connected component can be measured. We also consider the case of barcodes when no learning of the filtration occurs.

---

[1]https://github.com/simonzhang00/GraphExtendedFiltrationLearning

Zhang et al., GEFL: Extended Filtration Learning for Graph Classification. *Proceedings of the First Learning on Graphs Conference (LoG 2022)*, PMLR 198, Virtual Event, December 9–12, 2022.

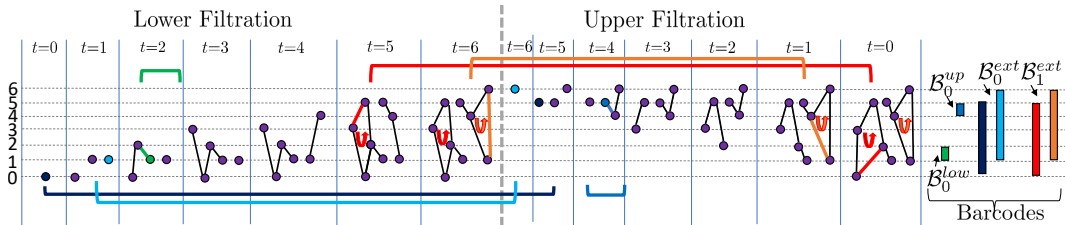

**Figure 1:** Lower and upper filtrations for extended persistence and the resulting barcode for a graph. The green bar comes from a pairing of a green edge with a vertex in the lower filtration. Similarily the blue bar in the upper filtration comes from a vertex-edge pairing in the upper filtration. The two dark blue bars count connected components and come from pairs of two vertices. The two red bars count cycles and come from pairs of edges. Both $\mathcal{B}_0^{ext}$ and $\mathcal{B}_1^{ext}$ bars cross from the lower filtration to the upper filtration. The multiset of bars forms the barcode. Cycle reps. are shown in both filtrations.

We consider several simple examples where our model can perfectly distinguish two classes of graphs that no GNN with expressivity at most that of WL[1] (henceforth called WL[1] bounded GNN) can. Furthermore, we present a case where experimentally 0-dimensional standard persistence [2, 5], the only kind of persistence considered in learning persistence so far, are insufficient for graph classification.

Our contributions are as follows:

1. We introduce extended persistence and its cycle representatives into the supervised learning framework in an end-to-end differentiable manner, for graph classification.

2. For a graph with $m$ edges and $n$ vertices, we introduce the link-cut tree data structure into the computation of extended persistence, resulting in an $O(m \log n)$ depth and $O(mn)$ work parallel algorithm, achieving more than 60x speedup over the state-of-the-art for extended persistence computation, making extended persistence amenable for machine learning tasks.

3. We analyze conditions and examples upon which extended persistence can surpass the WL[1] graph isomorphism test [6] and 0-dimensional standard persistence and characterize what extended persistence can measure from additional topological information.

4. We perform experiments to demonstrate the feasibility of our approach against standard baseline models and datasets as well as an ablation study on the readout function for a learned filtration.

## 2 Background

### 2.1 Computational Topology for Graphs

Let $G = (V, E)$ be a graph where $V$ is the set of vertices and $E \subset V \times V$ is the set of edges. Let $n = |V|$ and $m = |E|$ be the number of nodes and edges of $G$, respectively. Graphs in our case are undirected and simple, containing at most a single edge between any two vertices. Define a filtration function $F : G \to \mathbb{R}$ where $F$ has a value in $\mathbb{R}$ on each vertex and edge, denoted by $F(u)$ or $F(e)$ for $u \in V$ or $e \in E$. Given such a graph $G = (V, E)$, we define the $\lambda$-sublevel graph as $G_\lambda = (V_\lambda, E_\lambda)$ w.r.t. $F$ and a $\lambda \in \mathbb{R}$ where $V_\lambda = \{v \in V : F(v) \le \lambda\}$ and $E_\lambda = \{e \in E : F(e) \le \lambda\}$. Sublevel graphs of $G$ are subgraphs of $G$. If we change $\lambda$ from $-\infty$ to $+\infty$ we obtain an increasing sequence of sublevel graphs $\{G_\lambda\}_{\lambda \in \mathbb{R}}$ which we call a sublevel set filtration. Such a filtration can always be converted into a sequence of subgraphs of $G$: $\emptyset = G_0 \subset G_1 \subset ... \subset G_{n+m} = G$ (See [7, Page 102]) s.t. $\sigma_i = G_{i+1} \backslash G_i$ is a single edge or vertex and $F_i := F(\sigma_i)$. The sequence of vertices and edges $\sigma_0, \sigma_1, \ldots, \sigma_{n+m-1}$ thus obtained is called the index filtration. Define a vertex-induced lower filtration for a vertex function $f_G : V \to \mathbb{R}$ as an index filtration where a vertex $v$ has a value $F(v) := f_G(v)$ and any edge $(u, v)$ has the value $F(u, v) := \max(F(u), F(v))$ and $F_i \le F_{i+1}$. Similarly define an upper filtration for $f_G$ as an index filtration where $F(v) := f_G(v)$ and the edge $(u, v)$ has value $F(u, v) := \min(f_G(u), f_G(v))$ and $F_i \ge F_{i+1}$.

**Persistent homology**(PH) tracks changes in homological features of a topological space as the sublevel set for a given function grows; see books [7, 8]. For graphs, these features are given by evolution of components and cycles over the intervals determined by pairs of vertices and edges.

A vertex $v_i = G_{i+1} \setminus G_i$ begins a connected component (CC) signalling a birth at filtration value $F(v_i)$ in zeroth homology group $H_0$. An edge $e_j = G_{j+1} \setminus G_j$ may join two components signalling a death of a class in $H_0$ at filtration value $F(e_j)$, or it may create a cycle signalling a birth in the 1st homology group $H_1$ at filtration value $F(e_j)$. When a death occurs in $H_0$ by an edge $e_j$, the youngest of the two components being merged is said to die giving a birth-death pair $(b, d) = (F(v_i), F(e_j))$ if the dying component was created by vertex $v_i$. For cycles, there is no death and thus they have death at $\infty$. The multiset of birth death pairs $\mathcal{B} = \{\{(b, d)\}\}$ given by the persistent homology is called the barcode. Each pair $(b, d)$ provides a closed-open interval $[b, d)$, which is called a bar. The persistence of each bar $[b, d)$ in a barcode is defined as $|d - b|$. Notice that, both in 0- and 1-dimensional persistence, some bars may have infinite persistence since some components ($H_0$ features) and cycles ($H_1$ features) never die, equivalently, have death at $\infty$.

**Extended persistence($\mathbf{PH_{ext}}$)** takes an extended filtration $F_{f_G}$ as input, which is obtained by concatenating lower filtration of the graph $G$ and an upper filtration of the coned space of $G$ induced by a vertex filtration function $f_G$. Concatenation here simply means concatenating two index filtration sequences. More specifically, let $\alpha$ be an additional vertex for the graph $G$. Define an extended function $f_{G \cup \{\alpha\}}$ whose value is equal to $f_G$ on all vertices except $\alpha$ on which it has a value larger than any other vertices. The cone of a vertex $u$ is given by the edge $(\alpha, u)$ and the cone of an edge $(u, v)$ is given by the triangle $(\alpha, u, v)$. As a result, in extended persistence all 0- and 1-dimensional features die (bars are finite; see [3] for details). Four different persistence pairings or bars result from $\mathbf{PH_{ext}}$. The barcode $\mathcal{B}_0^{low}$ results from the vertex-edge pairs within the lower filtration, the barcode $\mathcal{B}_0^{up}$ results from the vertex-edge pairs within the upper filtration, the barcode $\mathcal{B}_0^{ext}$ results from the vertex-vertex pairs that represent the persistence of connected components born in the lower filtration and die in the upper filtration, and the barcode $\mathcal{B}_1^{ext}$ results from edge-edge pairs that represent the persistence of cycles that are born in the lower filtration and die in the upper filtration. The barcodes $\mathcal{B}_0^{low}$, $\mathcal{B}_0^{up}$, and $\mathcal{B}_0^{ext}$ represent persistence in the 0th homology $H_0$. The barcode $\mathcal{B}_1^{ext}$ represents persistence in the 1st homology $H_1$. In the TDA literature, $\mathcal{B}_0^{low}$, $\mathcal{B}_0^{up}$, $\mathcal{B}_0^{ext}$, and $\mathcal{B}_1^{ext}$ also go by the names of $Ord_0, Rel_1, Ext_0, Ext_1$ respectively.

See Figure 1 for an illustration of the filtration and barcode one obtains for a simple graph with vertices taking on values from $0...6$ denoted by the variable $t$. In particular, at each $t$, we have the filtration subgraph $G_t$ of all vertices and edges of filtration function value less than or equal to $t$. Each line indicates the values $0...6$ from the bottom to top. Repetition in the bar endpoints across all bars which appear on the right of Figure 1 is highly likely in general due to the fact that there are only $O(n)$ filtration values but $O(m)$ possible bars.

## 2.2 Message Passing Graph Neural Networks (MPGNN)

A message passing GNN (MPGNN) convolutional layer takes a vertex embedding $\mathbf{h}_u$ in a finite dimensional Euclidean space and an adjacency matrix $A_G$ as input and outputs a vertex embedding $\mathbf{h}_u'$ for some $u \in V$. The $k$th layer is defined generally as

$$\mathbf{h}_u^{k+1} \leftarrow \text{AGG}(\{\text{MSG}(\mathbf{h}_v^k) | v \in N_{A_G}(u)\}, \mathbf{h}_u^k), u \in V$$

where $N_{A_G}(u)$ is the neighborhood of $u$. The functions MSG and AGG have different implementations and depend on the type of GNN.

Since there should not be a canonical ordering to the nodes of a GNN in graph classification, a GNN for graph classification should be permutation invariant with respect to node indices. To achieve permutation invariance [9], as well as achieve a global view of the graph, there must exist a readout function or pooling layer in a GNN. The readout function is crucial to achieving power for graph classification. With a sufficiently powerful readout function, a simple 1-layer MPGNN with $O(\Delta)$ number of attributes [10] can compute any Turing computable function, $\Delta$ being the max degree of the graph. Examples of simple readout functions include aggregating the node embeddings, or taking the element-wise maximum of node embeddings [11]. See Section 3 for various message passing GNNs and readout functions from the literature.

## 3 Related Work

Graph Neural Networks (GNN)s have achieved state of the art performance on graph classification tasks in recent years. For a comprehensive introduction to GNNs, see the survey [12]. In terms of the

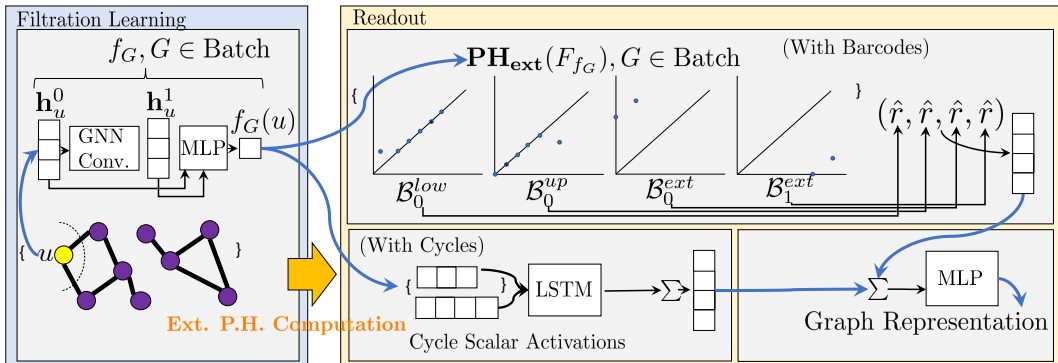

**Figure 2:** The extended persistence architecture (bars+cycles) for graph representation learning. The negative log likelihood (NLL) loss is used for supervised classification. The yellow arrow denotes extended persistence computation, which can compute both barcodes and cycle representatives.

Weisfeler Lehman (WL) hierarchy, there has been much success and efficiency in GNNs [11, 13, 14] bounded by the WL[1] [15] graph isomorphism test. In recent years, the WL[1] bound has been broken by heterogenous message passing [16], high order GNNs [17], and put into the framework of cellular message passing networks [18]. Furthermore, a sampling based pooling layer is designed in [19]. It has no theoretical guarantees and its code is not publicly available for comparison. Other readout functions include [20], [21] [22]. For a full survey on global pooling, see [23].

Topological Data Analysis (TDA) based methods [2, 5, 24–28] that use learning with persistent homology have achieved favorable performance with many conventional GNNs in recent years. All existing methods have been based on 0-dimensional standard persistent homology on separated lower and upper filtrations [5]. We sidestep these known limitations by introducing extended persistence into supervised learning while keeping computation efficient.

A TDA inspired cycle representation learning method in [29] learns the task of knowledge graph completion. It keeps track of cycle bases from shortest path trees and has a $O(|V| \cdot |E| \cdot k)$, $k$ a constant, computational complexity per graph. This high computational cost is addressed in our method by a more efficient algorithm for keeping track of a cycle basis. Furthermore, since the space of cycle bases induced by spanning forests is a strict subset of the space of all possible cycle bases, the extended persistence algorithm can find a cycle basis that the method in [29] cannot.

On the computational side, fast methods to compute higher dimensional PH using GPUs, a necessity for modern deep learning, have been introduced in [30]. In [27, 31] neural networks have been shown to successfully approximate the persistence diagrams with learning based approach. However, differentiability and parallel extended persistence computation has not been implemented. Given the expected future use of extended persistence in graph data, a parallel differentiable extended persistence algorithm is an advance on its own.

## 4 Method

Our method as illustrated in Figure 2 introduces extended persistence as the readout function for graph classification. In our method, an upper and lower filtration, represented by a filtration function, coincides with a set of scalar vertex representations from standard message passing GNNs. This filtration function is thus learnable by MPGNN convolutional layers. Learning filtrations was originally introduced in [5] with standard persistence. As we show in Section 6 and Section 5 arbitrary cycle lengths are hard to distinguish by both standard GNN readout functions [32] as well as standard persistence due to the lack of explicitly tracking paths or cycles. Extended persistence, on the other hand, explicitly computes learned displacements on cycles of some cycle basis as determined by the filtration function as well as explicit cycle representatives.

We represent the map from graphs to learnable filtrations by any message passing GNN layer such as GIN, GCN or GraphSAGE followed by a multi layer perceptron (MLP) as a Jumping Knowledge (JK) [33] layer. The JK layer with concatenation is used since we want to preserve the higher frequencies from the earlier layers [34]. Our experiments demonstrate that fewer MPGNN layers

perform better than more MPGNN layers. This prevents oversmoothing [35, 36], which is exacerbated by the necessity of scalar representations.

The readout function, the function that consolidates a filtration into a global graph representation, is determined by computing four types of bars for the extended persistence on the concatenation of the lower and upper filtrations followed by compositions with four rational hat functions $\hat{r}$ as used in [1, 2, 5]. To each of the four types of bars in barcode $\mathcal{B}$, we apply the hat function $\hat{r}$ to obtain a $k$-dimensional vector. The function $\hat{r}$ is defined as:

$$\hat{r}(\mathcal{B}) := \left\{ \sum_{\mathbf{p} \in \mathcal{B}} \frac{1}{1 + |\mathbf{p} - \mathbf{c_i}|_1} - \frac{1}{1 + ||r_i| - |\mathbf{p} - \mathbf{c_i}|_1|} \right\}_{i=1}^{k} \tag{1}$$

where $r_i \in \mathbb{R}$ and $\mathbf{c_i} \in \mathbb{R}^2$ are learnable parameters. The intent of Equation 1 is to have controlled gradients. It is derived from a monotonic function, see [1]. This representation is then passed through MLP layers followed by a softmax to obtain prediction probability vector $\hat{p}_G$ for each graph $G$. The negative log likelihood loss from standard graph classification is then used on these vectors $\hat{p}_G$.

If the filtration values on the nodes and edges are distinct, the extended persistence barcode representation is permutation invariant with respect to node indices. Isomorphic graphs with permuted indices and an index filtration with distinct filtration values will have a unique sorted index filtration. Node filtration values are usually distinct since computed floating points rarely coincide. However to break ties and eliminate any dependence on node indices for edges, implement edge filtration values for lower filtration as $F(u, v) = \max(F(u), F(v)) + \epsilon \cdot \min(F(u), F(v))$ and for upper filtration as $F(u, v) = \min(F(u), F(v)) + \epsilon \cdot \max(F(u), F(v))$, $\epsilon$ being very small.

**Cycle Representatives:** A cycle basis of a graph is a set of cycles where every cycle can be obtained from it by a symmetric difference, or sum, of cycles in the cycle basis. It can be shown that every cycle is a sum of the cycles induced by a spanning forest and the complementary edges. Extended persistence computes the same number of *independent* cycles as in the cycle basis induced from a spanning forest. Thus computing extended persistence results in computing a cycle basis. We can explicitly store the cycle representatives, or sequences of filtration scalars, along with the barcode on graph data. This slightly improves the performance in practice and guarantees cycle length classification for arbitrary lengths. After the cycle representatives are stored, we pass them through a bidirectional LSTM then aggregate these LSTM representation per graph and then sum this graph representation by cycles with the vectorization of the graph barcode by the rational hat function of Equation 1, see Figure 2. The aggregation of the cycle representations is permutation invariant due to the composition of aggregations [9]. In particular, the sum of the barcode vectorization and the mean of cycle representatives, our method's graph representation, must be permutation invariant. What makes keeping track of cycle representatives unique to standard message passing GNNs is that a finite receptive field message passing GNN would never be able to obtain such cycle representations and certainly not from a well formed cycle basis.

## 4.1 Efficient Computation of Extended Persistence

The computation for extended persistence can be reduced to applying a matrix reduction algorithm to a coned matrix as detailed in [8]. In [4], this computation was found to be equivalent to a graph algorithm, which we improve upon.

### 4.1.1 Algorithm

Our algorithm is as follows and written in Algorithm 1. We perform the $0$-dimensional persistence algorithm, $PH_0$, using the union find data structure in $O(m \log n)$ time and $O(n)$ memory for the upper and lower filtrations in lines 1 and 2. See the Appendix Section D.1 for a description of this algorithm. These two lines generate the vertex-edge pairs for $\mathcal{B}_0^{low}$ and $\mathcal{B}_0^{up}$. We then measure the minimum lower filtration value and maximum upper filtration value of each vertex in the union-find data structure found from the $PH_0$ algorithm as in lines 3 and 4 using the roots of the union-find data structure $U_{up}$ formed by the algorithm. These produce the vertex-vertex pairs in $\mathcal{B}_0^{ext}$.

For computing edge-edge pairs in $\mathcal{B}_1^{ext}$ with cycle representatives, we implement the algorithm in [4] with a link-cut tree data structure that facilitates deleting and inserting edges in a spanning tree and employ a parallel algorithm to enumerate the edges in a cycle. See the Appendix Section D.2

---

**Algorithm 1** Efficient Computation of $\mathbf{PH_{ext}}$

---

**Input:** $G = (V, E)$, $F_{low}$: lower filtration function, $F_{up}$: upper filtration function
**Output:** $\mathcal{B}_0^{low}, \mathcal{B}_0^{up}, \mathcal{B}_0^{ext}, \mathcal{B}_1^{ext}, \mathcal{C}$: cycle reps.
1: $\mathcal{B}_0^{low}, E_{pos}^{low}, E_{neg}^{low}, U_{low} \leftarrow \text{PH}_0(G, F_{low}, lower)$
2: $\mathcal{B}_0^{up}, E_{pos}^{up}, E_{neg}^{up}, U_{up} \leftarrow \text{PH}_0(G, F_{up}, upper)$
3: $roots \leftarrow \{\text{GET\_UNION-FIND\_ROOTS}(U_{up}, v), v \in V\}$
4: $\mathcal{B}_0^{ext} \leftarrow \{\min(roots[v]), \max(roots[v]), v \in V\}$
5: $\mathbf{T} \leftarrow \{\}$ empty link-cut tree; $\mathcal{B}_1^{ext} \leftarrow \{\{\}\}$; $\mathcal{C} \leftarrow \{\}$ empty list of cycle representatives
   /* $E_{neg}^{up}$ is sorted by $\text{PH}_0$ in decreasing order of $F_{up}$ values (desc. filtr. values)*/
6: **for** $e = (u, v) \in E_{neg}^{up}$ **do**
7:     $\mathbf{T} \leftarrow \text{LINK}(\mathbf{T}, e, \{w\})$ /* $w \notin \mathbf{T}$, $w = u$ or $v$*/
8: **end for**
9: /* $E_{pos}^{up}$ is sorted by $\text{PH}_0$ with respect to $F_{up}$ (descending filtration values) */
10: **for** $e = (u, v) \in E_{pos}^{up}$ **do**
11:     $lca \leftarrow \text{LCA}(\mathbf{T}, u, v)$ /*Get the least common ancestor of $u$ and $v$ to form a cycle*/
12:     $P_1 \leftarrow \text{LISTRANK}(\text{PATH}(u, lca))$; $P_2 \leftarrow \text{LISTRANK}(\text{PATH}(v, lca))$
13:     $\mathcal{C} \leftarrow \mathcal{C} \sqcup \{F_{up}(P_1) \sqcup F_{up}(Reverse(P_2))\}$ /*Keep track of the scalar activations on cycle*/
14:     $(u', v') \leftarrow \text{ARGMAXREDUCECYCLE}(\mathbf{T}, u, v, lca)$ /* Find max edge on cycle using $F_{low}$*/
15:     $\mathbf{T_1}, \mathbf{T_2} \leftarrow \text{CUT}(\mathbf{T}, (u', v'))$; $\mathbf{T} \leftarrow \text{LINK}(\mathbf{T_1}, (u, v), \mathbf{T_2})$
16:     $\mathcal{B}_1^{ext} \leftarrow \mathcal{B}_1^{ext} \cup \{(F_{low}(u', v'), F_{up}(u, v))\}$
17: **end for**
18: **return** $(\mathcal{B}_0^{low}, \mathcal{B}_0^{up}, \mathcal{B}_0^{ext}, \mathcal{B}_1^{ext}, \mathcal{C})$

---

for a more thorough explanation of the link-cut tree implementation and the operations we use on it. We collect the max spanning forest $T$ of negative edges, edges that join components, from the upper filtration by repeatedly applying the link operation $n - 1$ times in lines 6-8 in decreasing order of $F_{up}$ values and sort the list of the remaining positive edges, which create cycles in line 9. Then, for each positive edge $e = (u, v)$, in order of the upper filtration (line 10), we find the least common ancestor ($lca$) of $u$ and $v$ in the spanning forest $T$ we are maintaining as in line 11. Next, we apply the parallel primitive [37] of *list ranking* twice, once on the path $u$ to $lca$ and the other on the path $v$ to $lca$ in line 12. List ranking allows a list to populate an array in parallel in logarithmic time. The tensor concatenation of the two arrays is appended to a list of cycle representatives as in line 13. This is so that the cycle maintains order from $u$ to $v$. We then apply an ARGMAXREDUCECYCLE($T, u, v, lca$) which finds the edge having a maximum filtration value in the lower filtration on it over the cycle formed by $u, v$ and $lca$. We then cut the spanning forest at the edge $(u', v')$, forming two forests as in line 15. These two forests are then linked together at $(u, v)$ as in line 15. The bar $(F_{low}(u', v'), F_{up}(u, v))$ is now found and added to the multiset $\mathcal{B}_1^{ext}$. The final output of the algorithm is four types of bars and a list of cycle representatives: $((\mathcal{B}_0^{low}, \mathcal{B}_0^{up}, \mathcal{B}_0^{ext}, \mathcal{B}_1^{ext}), \mathcal{C})$.

### 4.1.2 Complexity

We improve upon the complexity of [4] by obtaining a $O(mn)$ work $O(m \log n)$ depth algorithm on $O(n)$ processors using $O(n)$ memory. Here $m$ and $n$ are the number of edges and vertices in the input graph. We introduce two ingredients for lowering the complexity, the first is the link-cut dynamic connectivity data structure and the second is the parallel primitives of list ranking. The link-cut tree data structure is a dynamic connectivity data structure that can keep track of the spanning forest with $O(\log n)$ amortized time for LINK, CUT, PATH, LCA, ARGMAXREDUCE. Furthermore, list ranking [38] is an $O(\log n)$ depth and $O(n)$ work parallel algorithm on $O(\frac{n}{\log n})$ processors that determines the distance of each vertex from the start of the path or linked list it is on. In other words, list ranking turns a linked list into an array in parallel. Sorting can be performed in parallel using $O(n \log n)$ work and $O(\log n)$ depth.

Notice that if we do not keep track of cycle representatives (remove lines 12 and 13 from Algorithm 1), then we have an $O(m \log n)$ time sequential algorithm. The repeated calling of the supporting operation EXPOSE() dominates the complexity, see Appendix Section D.2.

# 5 Expressivity of Extended Persistence

We prove some properties of extended persistence barcodes. We also find a case where extended persistence with supervised learning can give high performance for graph classification. WL[1] bounded GNNs, on the other hand, are guaranteed to not perform well. *Certainly all such results also apply for the explicit cycle representatives since the min and max on the scalar activations on the cycle form the corresponding bar.*

## 5.1 Some Properties

The following Theorem 5.1 states some properties of extended persistence. This should be compared with the 0- and 1-dimensional persistence barcodes in the standard persistence. Every vertex and edge is associated with some bar in the standard persistence though they can be both finite or infinite. However, in extended persistence all bars are finite and we form barcodes from an extended filtration of $2m + 2n$ edges and vertices instead of the standard $(m + n)$-lengthed filtration.

**Theorem 5.1.** *(Extended Barcode Properties)*

$\mathbf{PH_{ext}}(G)$ *produces four multisets of bars:* $\mathcal{B}_1^{ext}, \mathcal{B}_0^{ext}, \mathcal{B}_0^{low}, \mathcal{B}_0^{up}$, *s.t.*

$|\mathcal{B}_1^{ext}| = \dim H_1 = m - n + C,$

$|\mathcal{B}_0^{ext}| = \dim H_0 = C,$

$|\mathcal{B}_0^{low}| = |\mathcal{B}_0^{upper}| = n - C,$

*where there are $C$ connected components and $\dim H_k$ is the dimension of the kth homology group s.t.:*

*1. the $H_1$ bars comes from a cycle basis of $G$ which also constitutes a basis of its fundamental group,*

*2. $\dim H_1$ counts the number of chordless cycles when $G$ is outer-planar, and*

*3. there exists an injective filtration function where the union of the resulting barcodes is strictly more expressive than the histogram produced by the WL[1] graph isomorphism test.*

The barcodes found by extended persistence thus have more degrees of freedom than those obtained from standard persistence. For example, a cycle is now represented by two filtration values rather than just one. Furthermore, the persistence $|d - b|$ of a pair $(b, d) \in \mathcal{B}_1^{ext}$ or $\mathcal{B}_0^{ext}$ can measure topological significance of a cycle or a connected component respectively through persistence. Thus, extended persistence encodes more information than standard persistence. In Theorem 5.1, property 1 says that extended persistence actually computes pairs of edges of cycles in a cycle basis. A modification of the extended persistence algorithm could generate all or count certain kinds of important cycles, see [39]. Property 2 characterizes what extended persistence can count.

We makes some observations on the expressivity of $\mathbf{PH_{ext}}$.

**Observation 5.2.** (Cycle Lengths) For any graph $G$ and chordless cycle $\mathbf{C} \subset G$, there exists injective filtration functions $f_G^{low}, f_G^{up}$ on $G$ where $\mathbf{PH_{ext}}$ of the induced filtration for extended persistence can measure the number of edges along $\mathbf{C}$.

Such a result cannot hold for learning of the filtration by local message passing from constant node attributes. Thus, for the challenging 2CYCLE graphs dataset in Section B.2, it is a necessity to use the cycle representatives $\mathcal{C}$ for each graph to distinguish pairs of cycles of arbitrary length. This should be compared with Top-$K$ methods, $K$ being a constant hyper parameter such as in [19, 40]. The constant hyper parameter $K$ prevents learning an arbitrarily long cycle length when the node attributes are all the same. Furthermore, a readout function like SUM is agnostic to graph topology and also struggles with learning when presented with an arbitrarily long cycle. This struggle for distinguishing cycles in standard MPGNNs is also reported in [41]. An observation similar to the previous Observation 5.2 can also be made for paths measured by $\mathcal{B}_0^{ext}$.

**Observation 5.3.** (Connected Component Sizes) For any graph $G$ and all connected components $\mathbf{CC} \subset G$, there exists injective filtration functions $f_G^{low}, f_G^{up}$ defined on $G$ where $\mathbf{PH_{ext}}$ of the induced filtration for extended persistence can measure the number of vertices in $\mathbf{CC}$.

We investigate the case where no learning takes place, namely when the filtration values come from a random noise. We observe that even in such a situation some information is still encoded in the extended persistence barcodes with a probability that depends on the graph.

**Observation 5.4.** For any graph $G$ where every edge belongs to some cycle and an extended filtration on it induced by randomly sampled vertex values $x_i \sim U([0,1])$, $\mathbf{PH_{ext}}$ has a $H_1$ bar $[max_i(x_i), min_i(x_i)]$ with probability $\sum_{v \in V} \frac{1}{n} \frac{deg(v)}{n-1} = \frac{2m}{n(n-1)}$.

Notice that for a clique, the probability of finding the bar with maximum possible persistence is 1. It becomes lower for sparser graphs.

**Corollary 5.5.** *In Observation 5.4, assuming the bar $[max_i(x_i), min_i(x_i)]$ exists, the expected persistence of that bar $\mathbb{E}[|max_i(x_i) - min_i(x_i)|]$ goes to 1 as $n \to \infty$.*

What Corollary 5.5 implies is that, for certain graphs, even when nothing is learned by the GNN filtration learning layers, the longest $\mathcal{B}_1^{ext}$ bar indicates that $n$ is large. This happens for graphs that are randomly initialized with vertex labels from the unit interval and occurs with high probability for dense graphs by Observation 5.4. For large $n$, the empirical mean of the longest bar will have persistence near 1. Notice that $\mathcal{B}_1^{ext}$ can measure this even though the number of $H_1$ bars, $m - n + C$, could tell us nothing about $n$.

## 6  Experiments

We perform experiments of our method on standard GNN datasets. We also perform timing experiments for our extended persistence algorithm, showing impressive scaling. Finally, we investigate cases where experimentally our method distinguishes graphs that other methods cannot, demonstrating how our method learns to surpass the WL[1] bound.

### 6.1  Experimental Setup

We perform experiments on a 48 core Intel Xeon Gold CPU machine with 1 TB DRAM equipped with a Quadro RTX 6000 NVIDIA GPU with 24 GB of GPU DRAM.

Hyper parameter information can be found in Table 3. For all baseline comparisons, the hyperparameters were set to their repository's standard values. In particular, all training were stopped at 100 epochs using a learning rate of 0.01 with the Adam optimizer. Vertex attributes were used along with vertex degree information as initial vertex labels if offered by the dataset. We perform a fair performance evaluation by performing standard 10-fold cross validation on our datasets. The lowest validation loss is used to determined a test score on a test partition. An average±standard deviation test score over all partitions determines the final evaluation score.

The specific layers of our architecture for the neural network for our filtration function $f_G$ is given by one or two GIN convolutional layers, with the number of layers as determined by an ablation study.

### 6.2  Performance on Real World and Synthetic Datasets

We perform experiments with the TUDatasets [42], a standard GNN benchmark. We compare with WL[1] bounded GNNs (GIN, GIN0, GraphSAGE, GCN) from the PyTorch Geometric [43, 44] benchmark baseline commonly used in practice as well as GFL[5], ADGCL [45], and InfoGraph [46], self-supervised methods. Self supervised methods are promising but should not surpass the performance of supervised methods since they do not use the label during representation learning. We also compare with existing topology based methods TOGL [2] and GFL [5]. We also perform an ablation study on the readout function, comparing extended persistence as the readout function with the SUM, AVERAGE, MAX, SORT, and SET2SET [47] readout functions. The hyper parameter $k$ is set to the 10th percentile of all datasets when sorting for the top-$k$ nodes activations. We do not compare with [19] since its code is not available online. The performance numbers are listed in Table 1. We are able to improve upon other approaches for almost all cases. The real world datasets include DD, MUTAG, PROTEINS and IMDB-MULTI. DD, PROTEINS, and MUTAG are molecular biology datasets, which emphasize cycles, while IMDB-MULTI is a social network, which emphasize cliques and their connections. We use accuracy as our performance score since it is the standard for the TU datasets.

We also verify that our method surpasses the WL[1] bound, a theoretical property which can be proven, as well as can count cycle lengths when the graph is sparse enough, e.g. when the set of cycles is equal to the cycle basis. This is achieved by the two datasets PINWHEELS and 2CYCLES. See the Appendix Sections B for the related experimental and dataset details. Both datasets are

| Experimental Evaluation | | | | | | |
|---|---|---|---|---|---|---|
| avg. acc. $\pm$ std. | DD | PROTEINS | IMDB-MULTI | MUTAG | PINWHEELS | 2CYCLES |
| GFL | 75.2 ± 3.5 | 73.0 ± 3.0 | 46.7 ± 5.0 | 87.2 ± 4.6 | 100 ±0.0 | 50.0 ±0.0 |
| **Ours+Bars** | 75.5 ± 2.9 | 74.9 ± 4.1 | 50.3 ± 4.7 | **88.3 ±7.1** | **100 ±0.0** | 50 ± 0.0 |
| **Ours+bars+cycles** | 75.9 ± 2.0 | **75.2 ± 4.1** | **51.0 ± 4.6** | 86.8 ± 7.1 | **100 ±0.0** | **100 ± 0.0** |
| GIN | 72.6± 4.2 | 66.5 ± 3.8 | 49.8 ± 3.0 | 84.6 ± 7.9 | 50.0 ±0.0 | 50.0 ±0.0 |
| GIN0 | 72.3 ± 3.6 | 67.5 ± 4.7 | 48.7 ± 3.7 | 83.5 ± 7.4 | 50.0 ±0.0 | 50.0 ±0.0 |
| GraphSAGE | 72.6 ± 3.7 | 59.6 ± 0.2 | 50.0 ± 3.0 | 72.4 ± 8.1 | 50.0 ±0.0 | 50.0 ±0.0 |
| GCN | 72.7 ± 1.6 | 59.6 ± 0.2 | 50.0 ± 2.0 | 73.9 ± 9.3 | 50.0 ±0.0 | 50.0 ±0.0 |
| GraphCL | 65.4 ±12 | 62.5 ± 1.5 | 49.6± 0.4 | 76.6 ± 26 | 49.0 ±8.0 | 50.5± 10 |
| InfoGraph | 61.5 ± 10 | 65.5 ± 12 | 40.0 ± 8.9 | 89.1 ± 1.0 | 50.0 ± 0.0 | 50.0 ± 0.0 |
| ADGCL | 74.8± 0.7 | 73.2± 0.3 | 47.4 ± 0.8 | 63.3± 31 | 42.5 ± 19 | 52.5 ± 21 |
| TOGL | 74.7 ± 2.4 | 66.5 ± 2.5 | 44.7 ± 6.5 | - | 47.0 ± 3 | 54.4 ± 5.8 |
| Filt.+SUM | 75.0 ± 3.2 | 73.5 ± 2.8 | 48.0 ± 2.9 | 86.7± 8.0 | 51.0 ± 11 | 50.0 ± 0.0 |
| Filt.+MAX | 67.6± 3.9 | 68.6± 4.3 | 45.5 ± 3.1 | 70.3± 5.4 | 48.0 ± 4.2 | 50.0 ± 0.0 |
| Filt.+AVG | 69.5± 2.9 | 67.2± 4.2 | 46.7 ± 3.8 | 81.4± 7.9 | 50.0 ± 13 | 50.0 ± 0.0 |
| Filt.+SORT | **76.9± 2.6** | 72.6 ± 4.6 | 49.0± 3.6 | 85.6± 9.2 | 51.0 ± 16 | 50.0 ± 0.0 |
| Filt.+S2S | 69.0 ± 3.3 | 67.8 ± 4.6 | 48.7 ± 4.2 | 86.8 ± 7.1 | 51.0 ± 13 | 50.0 ± 0.0 |

**Table 1:** Average accuracy $\pm$ std. dev. of our approach (GEFL) with and without explicit cycle representations, Graph Filtration Learning (GFL), GIN0, GIN, GraphSAGE, GCN, ADGCL, GraphCL and TOGL and a readout ablation study on the four TUDatasets: DD, PROTEINS, IMDB-MULTI, MUTAG as well as the two Synthetic WL[1] bound and Cycle length distinguishing datasets. Numbers in bold are highest in performance; bold-gray numbers show the second highest. The symbol $-$ denotes that the dataset was not compatible with software at the time.

| 10-fold cross validation ablation study on OGBG-MOL datasets by ROC-AUC | | | | | | |
|---|---|---|---|---|---|---|
| avg. score $\pm$ std. | **Ours+Bars** | **Ours+Bars +Cycles** | Filt.+SUM | Filt.+MAX | Filt.+AVG | Filt.+SORT | Filt.+Set2Set |
| molbace | 80.0 ± 3.6 | **81.6 ± 3.9** | 79.7 ± 4.6 | 71.9 ± 4.8 | 78.0 ± 3.0 | 78.4 ± 3.3 | 78.2 ± 3.6 |
| molbbbp | 78.0 ± 4.3 | **81.9 ± 3.3** | 76.7 ± 4.9 | 69.8 ± 8.7 | 78.5 ± 4.6 | 76.3 ± 4.3 | 78.0 ± 5.0 |

**Table 2:** Ablation study on readout functions. The average ROC-AUC $\pm$ std. dev. on the ogbg-mol datasets is shown for each readout function. Number coloring is as in Table 1

particularly hard to classify since they contain spurious constant node attributes, with the labels depending completely on the graph connectivity. This removal of node attributes is in simulation of the WL[1] graph isomorphism test, see [6]. Furthermore, doing so is a case considered in [48]. It is known that WL[1], in particular WL[2], cannot determine the existence of cycles of length greater than seven [49, 50].

Table 2 shows the ablation study of extended filtration learning on the ogbg datasets [51] OGBG MOLBACE and MOLBBBP. We perform a 10 fold cross validation with the test ROC-AUC score of the lowest validation loss used as the test score. This is performed instead of using the train/val/test split offered by the OGBG dataset in order to keep our evaluation methods consistent with the evaluation of the TUDATASETS and synthetic datasets.

From Section B, we know that there are special cases where extended persistence can distinguish graphs where WL[1] bounded GNNs cannot. We perform experiments to show that our method can surpass random guessing whereas other methods achieve only $\sim 50\%$ accuracy on average, which is no better than random guessing. Our high accuracy is guaranteed on PINWHEELS since such graphs are distinguished by counting bars through 0-dim standard persistence. Similarly, 2CYCLES is guaranteed high accuracy when keeping track of cycles and comparing the variance of cycle representations since cycle lengths can be distinguished by a LSTM on different lengthed cycle inputs. Of course, a barcode representation alone will not distinguish cycle lengths.

# 7 Conclusion

We introduce extended persistence into the supervised learning framework, bringing in crucial global connected component and cycle measurement information into the graph representations. We address a fundamental limitation of MPGNNs, which is their inability to measure cycles lengths. Our method hinges on an efficient algorithm for computing extended persistence. This is a parallel differentiable algorithm with an $O(m \log n)$ depth $O(mn)$ work complexity and scales impressively over the state-of-the-art. The speed with which we can compute extended persistence makes it feasible for machine learning. Our end-to-end model obtains favorable performance on real world datasets. We also construct cases where our method can distinguish graphs that existing methods struggle with.

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

# A Proofs

**Theorem A.1.** *(Theorem 5.1)*

$\mathbf{PH_{ext}}(G)$ *produces four types of bars:* $\mathcal{B}_1^{ext}, \mathcal{B}_0^{ext}, \mathcal{B}_0^{low}, \mathcal{B}_0^{up}$, *s.t.*

$|\mathcal{B}_1^{ext}| = \dim H_1 = m - n + C$,

$|\mathcal{B}_0^{ext}| = \dim H_0 = C$,

$|\mathcal{B}_0^{low}| = |\mathcal{B}_0^{upper}| = n - C$,

*where there are $C$ connected components and $\dim H_k$ is the dimension of the kth homology group s.t.:*

*1. the $H_1$ barcode comes from a cycle basis of $G$ which also constitutes a basis of its fundamental group,*

*2. $\dim H_1$ counts the number of chordless cycles when $G$ is outer-planar, and*

*3. there exists an injective filtration function where the union of the resulting barcodes is strictly more expressive than the histogram produced by the WL[1] graph isomorphism test.*

*Proof.* There are $n$ bars with vertex births since every vertex creates exactly one connected component. The number of these bars which are in $\mathcal{B}_0^{ext}$ is $C$, which counts the number of global connected components. In other words, $\mathcal{B}_0^{ext} = \dim(H_0) = C$. Thus, we have $n - C = |\mathcal{B}_0^{low}| = |\mathcal{B}_0^{upper}|$.

Considering all $2m$ edges on the extended filtration, every edge gets paired. Furthermore, $n - C$ of the edges in the lower filtration are negative edges paired with vertices that give birth to connected components. Similarly there are $n - C$ edges paired with vertices in the upper filtration. We thus have $\frac{2m - 2(n-C)}{2}$ edge-edge pairings in $\mathcal{B}_1^{ext}$ because every edge gets paired. Thus, $|\mathcal{B}_1^{ext}| = m - n + C$. Since each bar in $\mathcal{B}_1^{ext}$ counts a birth of a 1-dimensional homological class which together span the 1-dimensional homological classes in $H_1$, we have that $\dim H_1 = |\mathcal{B}_1^{ext}|$.

1. All cycle representatives found by the algorithm are a symmetric difference of cycles from a *fundamental* cycle basis, or cycle basis induced from a spanning forest. This follows since the link and cut operations in our algorithm correspond to cycle additions in extended persistence computation. Furthermore, each cycle in the returned cycle representatives is independent because it has a unique cycle from the fundamental cycle basis as a summand. Also, there are $m - n + 1$ returned cycle representatives, same as the fundamental cycle basis. Thus the cycle representatives form a cycle basis.

Any cycle basis generates the fundamental group and $H_1(G, \mathbb{Z}_2)$ homology group of graph $G$ [52]

2. By Euler's formula, we have $n - m + F = C + 1$ for planar graphs where $F$ is the number of faces of the planar graph as embedded in $\mathbb{S}^2$. For outer planar graphs, since $F - 1$ interior faces lie on one hemisphere of $\mathbb{S}^2$ and one exterior face covers the opposite hemisphere, each interior face must be a chordless cycle.

3. This follows directly by the result in [2] stating that 0-dimensional barcodes are more expressive than the WL[1] graph isomorphism test. In extended persistence, $\mathcal{B}_0^{low}$ and $\mathcal{B}_0^{ext}$ are computed. Since all bars in $\mathcal{B}_0^{ext}$ correspond to infinite bars denoted $\mathcal{B}_0^{\infty}$ in the 0-dimensional standard persistence, we have that $\mathcal{B}_0^{low}$ and $\mathcal{B}_0^{ext}$ carry at least the same amount of information as a 0-dimensional barcode as determined by $\mathcal{B}_0^{low}$ and $\mathcal{B}_0^{\infty}$.

□

**Observation A.2.** (Observation 5.2) For any graph $G$ and chordless cycle $\mathbf{C} \subset G$, there exists injective filtration functions $f_G^{low}, f_G^{up}$ on $G$ where $\mathbf{PH_{ext}}$ of the induced filtration for extended persistence can measure the number of edges along $\mathbf{C}$.

*Proof.* Number the vertices of the cycle $\mathbf{C}$ of length $k$ in descending order and counter clockwise as $n - 1 \dots n - k$. For each vertex $u \in \mathbf{C}$, set $f_G^{low}(u) = f_G^{up}(u)$ to be the index of $u$ in the vertex numbering, the lower and upper filtration value on the nodes. For the other vertices, assign arbitrary different values less than $n - k$. The edge values are then assigned $f_G^{up}(u, v) = \min(f_G^{up}(u), f_G^{up}(v))$ and $f_G^{low}(u, v) = \max(f_G^{low}(u), f_G^{low}(v))$. Apply $\varepsilon$-perturbation to make the filtration functions

$f_G^{low}, f_G^{up}$ injective and to force exactly one edge on the cycle to be positive. Every edge is either positive or negative and all negative edges are on a spanning forest.

In particular, for vertex $u$ and all its incident edges of same upper filtration value, one can subtract different $\varepsilon \in \mathbb{R}^+$ from each edge to impose an order amongst edges with the same value from $f_G^{up}$. Similarly, for $f_G^{low}$ subtract different $\varepsilon \in \mathbb{R}^+$ to each edge to break ties. For the edges on $\mathbf{C}$, do not subtract an $\varepsilon$ in order to force each node $i$ on $\mathbf{C}$ to pair with the largest edge: $(i-1, i)$ of filtration value $i$. Since there is a tie for the edges in $\mathcal{C}$ incident to node $n - k$ in the upper filtration, set the edge $f_G^{up}(n-k, n-k+1) := n-k+\varepsilon$. This ensures that edge $(n-k, n-k+1)$ is negative and $(n-1, n-k)$ is positive in the upper filtration since the edges with larger filtration values are paired, or made negative, first. Similarly, there is a tie for the edges in $\mathcal{C}$ incident to node $n - 1$ in the lower filtration. Set $f_G^{low}(n-1, n-k) := n-1+\varepsilon$. This ensures that edge $(n-1, n-2)$ is positive in the lower filtration and $(n-1, n-k)$ is negative in the lower filtration. The resulting filtration functions $f_G^{low}$ and $f_G^{up}$ are injective. Furthermore, we then get that every edge on the cycle $\mathbf{C}$ except one: $(n-1, n-k)$, a positive edge, becomes negative in the upper filtration and thus belongs to the negative spanning forest of the upper filtration. The positive edge of smallest value in the upper filtration is edge $(n-1, n-k)$. The extended persistence algorithm, after computing $\mathcal{B}_0^{low}$ and $\mathcal{B}_0^{up}$, pairs the edge $e = (n-1, n-k)$ with the edge having maximum value in the lower filtration in the cycle $\mathbf{C}$ that $e$ forms with the spanning forest. This paired edge is $(n-1, n-2)$ and has lower filtration value $n-1$. We thus have the bar $[n-1, n-k]$ which encodes the length $k$ of the cycle $\mathbf{C}$.

$\square$

**Observation A.3.** (Observation 5.3) For any graph $G$ and all connected components $\mathbf{CC} \subset G$, there exists injective filtration functions $f_G^{low}, f_G^{up}$ defined on $G$ where $\mathbf{PH_{ext}}$ of the induced filtration for extended persistence can measure the number of vertices in $\mathbf{CC}$.

*Proof.* For each connected component $\mathbf{CC}$ in $G$, index the vertices in $\mathbf{CC}$ in consecutive order where indices in each connected component remain distinct. Then define $f_G^{low}(u) = f_G^{up}(u)$ equal to the index of $u$ in $G$. By some $\varepsilon$-perturbation, where we break ties amongst edges, we can make these two functions injective on the graph $G$. Since $\mathcal{B}_0^{ext}$ has each bar $[\min_{u \in \mathbf{CC}} f_G^{low}(u), \max_{u \in \mathbf{CC}} f_G^{up}(u)]$ and since all indices are consecutive, each bar's persistence in $\mathcal{B}_0^{ext}$ measures how many vertices are in the connected component they constitute.

$\square$

**Observation A.4.** (Observation 5.4) For any graph $G$ where every edge belongs to some cycle and an extended filtration on it is induced by randomly sampling vertex values $x_i \sim U([0,1])$, $\mathbf{PH_{ext}}$ has the $H_1$ bar $[\max_i(x_i), \min_i(x_i)]$ with probability $\sum_{v \in V} \frac{1}{n} \frac{deg(v)}{n-1} = \frac{2m}{n(n-1)}$.

*Proof.* Since the probability of finding a given permutation on $n$ vertices sampled uniformly at random without replacement is equivalent to the probability of a given order on the vertices sampled uniformly at randomly $n$ times, it suffices to find the probability of sampling uniformly at random without replacement two vertices that are connected with an edge in $G$.

For a fixed $\sigma \in S_n$, a permutation from the group $S_n$ of permutations on $n$ vertices, we have:

$$\frac{1}{n!} = P(x_n < x_{n-1} < ... < x_1, x_i \sim U([0,1]))$$
$$= \int_0^1 \int_0^{x_1} ... \int_0^{x_{n-1}} dx_n dx_{n-1}...dx_1 = P(\sigma \sim U(S_n)) \qquad (2)$$

Let $G = (V, E)$ be the graph with vertex values sampled from a uniform distribution. Let $G' = (V', E')$ be the same graph with vertex values in $\{0, 1, \ldots, n-1\}$ sampled uniformly without replacement. We know that the probability for a given order on these vertices is the same for both graphs. In fact, the two node labelings are in bijection with each other. By the law of total probability and with Equation 2:

$$P((\min_i x_i, \max_i x_i) \in E, x_i \sim U([0,1]))$$

$$= \sum_{v \in V} (P(v = \max_i x_i, x_i \sim U([0,1])) \cdot P(\min_i x_i \in Nbr(v) | v = \max_i x_i, x_i \sim U([0,1])))$$

$$= \sum_{v \in V} (n-1)! \int_0^1 \int_0^{x_1} ... \int_0^{x_{n-1}} dx_n dx_{n-1}...dx_1 \cdot deg(v)(n-2)! \int_0^1 \int_0^{x_2} ... \int_0^{x_{n-1}} dx_n dx_{n-1}...dx_2$$

$$= \sum_{v \in V'} (P(v = n-1) \cdot P(0 \in Nbr(v) | v = n-1)) = P((n-1, 0) \in E')$$

$$= \sum_{v \in V'} \frac{1}{n} \frac{deg(v)}{n-1}$$

We now show that if $(\min_i x_i, \max_i x_i)$ occurs as an edge in $G = (V, E)$, where every edge belongs to some cycle, then the bar $[\max_i x_i, \min_i x_i]$ is guaranteed to occur.

The spanning tree comprised of negative edges that begins the computation for $\mathcal{B}_1^{ext}$ as in line 6 of Algorithm 1 for the $H_1$ barcode computation is a maximum spanning tree. This is because the negative edges are just those found by the Kruskal's algorithm for the 0-dimensional standard persistence applied to an upper filtration. Since $e = (\min_i x_i, \max_i x_i)$ has value $\min_i x_i$ in the upper filtration and since every edge belongs to at least one cycle, it cannot be in the maximum spanning tree. Thus $e$ is a positive edge.

Since $e$ is positive in the upper filtration, it will be considered at some iteration of the for loop in line 10 of Algorithm 1. When we consider it, it will form a cycle $\mathbf{C}$ with the dynamically maintained spanning forest. To form a persistence $H_1$ bar for $e$, we pair it with the maximum edge in the cycle $\mathbf{C}$ in the lower filtration. This forms a bar $[\max_i x_i, \min_i x_i]$.

$\square$

**Corollary A.5.** *In Observation 5.4, assuming the bar $[max_i(x_i), min_i(x_i)]$ exists, the expected persistence of that bar, $\mathbb{E}[|max_i(x_i) - min_i(x_i)|]$, goes to 1 as $n \to \infty$.*

*Proof.* Define the random variable $X_n = |\max_i x_i - \min_i x_i|$ for $n$ random points drawn uniformly from $[0,1]$. We find $\lim_{n \to \infty} \mathbb{E}[X_n]$. The following sequence of equations follow by repeated substitution.

$$\mathbb{E}[X_n] = n! \int_0^1 \int_0^{x_1} ... \int_0^{x_{n-1}} (x_1 - x_n) dx_n...dx_1$$

$$= n! \int_0^1 (\frac{x_1^n}{(n-1)!} - \frac{x_1^n}{n!}) dx_1 = \frac{n-1}{n+1}$$

where the $n!$ comes from symmetry.

Therefore: $\lim_{n \to \infty} \mathbb{E}[X_n] = 1$.

$\square$

# B  Demonstrating the Expressivity of Learned Extended Persistence

We present some cases where the classification performance of our method excels. We look for graphs that cannot be distinguished by WL[1] bounded GNNs. We find that pinwheeled cycle graphs and varied length cycle graphs can be perfectly distinguished by learned extended persistence and, in practice, with much better performance than random guessing using our model. See the experiments Section 6 to see the empirical results for our method against other methods on this synthetic data.

## B.1  Pinwheeled Cycle Graphs (The PINWHEELS Dataset)

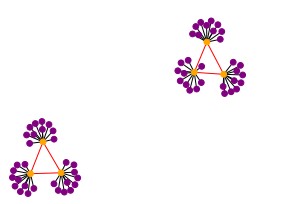

**Figure 3:** Class 0: 2 triangles with pinwheel at each vertex.

**Figure 4:** Class 1: A hexagon with pinwheel at each vertex.

We consider pinwheeled cycle graphs. To form the base skeleton of these graphs, we take the standard counter example to the WL[1] test of 2 triangles and 1 hexagon. We then append pinwheels of a constant number of vertices to the vertices of these base skeletons. The node attributes are set to a spurious constant noise vector. They have no effect on the labels.

It is easy to check that both Class 0 and Class 1 graphs are indistinguishable by WL[1]; see Figures 3 and 4. Notice that if there are 6 core vertices and edges in the base skeleton and if there are pinwheels of size $k$, then with edge deletions and vertex deletions composed, we have a $1 - (\frac{6}{6k+6})^2$ probability of only deleting a pinwheel edge or vertex and thus not affecting $H_1$. This probability converges to 1 as $k \to \infty$. According to Theorem 5.1, $\dim H_1$ measures the number of cycles and $\dim H_0$ measures the number of connected components. If neither of these counts are affected by training during supervised learning, our method is guaranteed to distinguish the two classes simply by counting according to Theorem 5.1.

Certainly the pinwheeled cycle graphs, are distinguishable by counts of bars. We check this experimentally by constructing a dataset of 1000 graphs of two classes of graph evenly split. Class 0 is as in Figure 3 and involves two triangles with pinwheels of random sizes. Class 1 is as in Figure 4 with a hexagon and pinwheels of random sizes attached. We obtain on average 100% accuracy. This is confirmed experimentally in Table 1. This matches the performance of GFL [5], since counting bars, or Betti numbers, can also be done through 0-dim. standard persistence. Interestingly TOGL does not achieve a score of 100 accuracy on this dataset. We conjecture this is because their layers are not able to ignore the spurious and in fact misleading constant node attributes.

## B.2  Regular Varied Length Cycle Graphs (The 2CYCLES dataset)

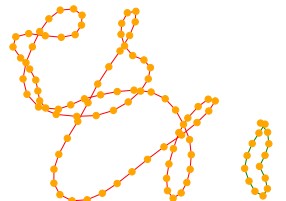

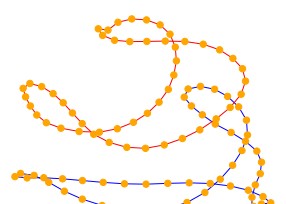

**Figure 5:** Class 0: A 15 node cycle and an 85 node cycle.

**Figure 6:** Class 1: A 50 node cycle with a 50 node cycle.

We further consider varied length cycle graphs. These are graphs that involve two cycles. Class 0 has one short and one long cycle while Class 1 has two near even lengthed cycles. The node attributes are all the same and spurious in this dataset. Extended persistence should do well to distinguish these

two classes. We conjecture this based on Observation 5.2, which states that there is some filtration that can measure the length of certain cycles.

It is the path length, coming from Observation 5.3, which is being measured. The 0-dimensional standard persistence is insufficient for this purpose. The infinite bars of 0-dimensional standard persistence are determined only by a birth time. Furthermore, extended persistence without cycle representatives is also insufficient since a message passing GNN learns a constant filtration function over the nodes. However, with cycle representatives, or a list of scalar node activations per cycle for each graph, we can easily distinguish the average sequence representation since the pair of sequence lengths are different. In class 0, a short cycle and a long cycle are paired while in class 1, two cycles of medium lengths are paired.

A similar but more challenging dataset to the PINWHEELS dataset, the 2CYCLES dataset, is similar to the necklaces dataset from [2] and is illustrated in Section B.2 but with more misleading node attributes and simplified to two cycles. It involves 400 graphs consisting of two cycles. There are two classes as shown in Figures 5 and 6.

The experimental performance on 2CYCLES surpasses random guessing while all other methods just randomly guess as stated in Section B.2. Certainly WL[1] bounded GNNs cannot distinguish the two classes in 2CYCLES since they are all regular. As discussed, because GFL and TOGL use learned 0-dimensional standard persistence, these approaches do no better than random guessing on this dataset.

## C   Timing of Extended Persistence Algorithm (without storing cycle representations)

Since the persistence computation, especially extended persistence computation, is the bottleneck to any machine learning algorithm that uses it, it is imperative to have a fast algorithm to compute it. We perform timing experiments with a C++ torch implementation of our fast extended persistence algorithm. In our implementation each graph in the batch has a single thread assigned to it.

Our experiment involves two parameters, the sparsity, or probability, $p$ for the edges of an Erdos-Renyi graph and the number of vertices of such a graph $n$. We plot our speedup over GUDHI, the state of the art software for computing extended persistence, as a function of $p$ with $n$ held fixed. We run GUDHI and our algorithm 5 times and take the average and standard deviation of each run's speedup. Since our algorithm has lower complexity, our speedup is theoretically unbounded. We obtain up to 62x speedup before surpassing 12 hours of computation time for experimentation. The plot is shown in Figure 7. The speedup is up to 2.8x, 9x, 24x, and 62x for $n = 200, 500, 1000, 2000$ respectively.

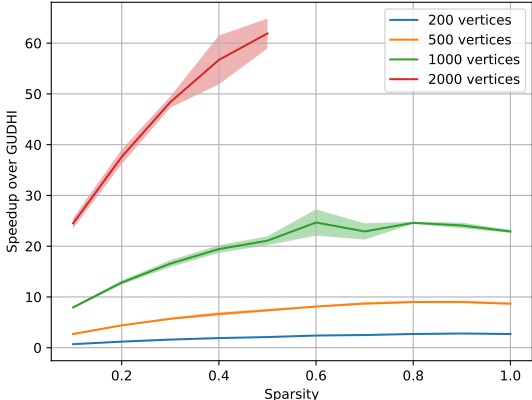

**Figure 7:** Average speedup with std. dev. as a function of sparsity $p$ and number of vertices $n$ on Erdos Renyi graphs.

## D   Algorithm and Data Structure Details

Here we detail the algorithmic details of computing extended persistence.

### D.1   The $PH_0$ Algorithm

Algorithm 2 is the union-find algorithm that computes 0 dimensional persistent homology. The algorithm is a single-linkage clustering algorithm [53]. It starts with $n$ nodes, 0 edges, and a union-find data structure [54] on $n$ nodes. The edges are sorted in ascending order if a lower filtration function is given. Otherwise, the edges are sorted in descending order. It then proceeds to connect nearest neighbor clusters, or connected components, in a sequential fashion by introducing edges in order one at a time. Two connected components are nearest to each other if they have two nodes closer to each other than any other pair of connected components. This is achieved by iterating through the edges in sorted order and merging the connected components that they connect. When given a lower filtration function, when a connected component merges with another connected component, the connected component with the larger connected component root value has its root filtration function value a birth time. This birth time is paired with the current edge's filtration value and form a birth death pair. The smaller of the two connected component root values is used as birth time when an upper filtration function is given. The two connected components are subsequently merged in a union-find data structure by the LINK operation.

---

**Algorithm 2** $PH_0$ Algorithm

---

    **Input:** $G = (V, E)$, $F$: filtration function, $order$: flag to denote an upper or lower filtration
    **Output:** $\mathcal{B}_0, E_{pos}, E_{neg}, U$: $H_0$ bars, pos. edges, neg. edges, and union-find data structure
1:  $U \leftarrow V$ /* a union-find data structure populated by n unlinked nodes*/
2:  $\mathcal{B}_0 \leftarrow \{\}$ /*A multiset */
3:  **if** $order = lower$ **then**
4:     $\text{SORT}_{incr}(E)$ /*increasing w.r.t. F;*/
5:  **else**
6:     $\text{SORT}_{decr}(E)$ /*decreasing w.r.t F;*/
7:  **end if**
8:  **for** $e = (u, v) \in E$ **do**
9:     $root_u \leftarrow U.\text{FIND}(u)$
10:    $root_v \leftarrow U.\text{FIND}(v)$
11:    **if** $root_u = root_v$ **then**
12:      $E_{pos} \leftarrow E_{pos} \cup \{e\}$
13:    **else**
14:      $E_{neg} \leftarrow E_{neg} \cup \{e\}$
15:    **end if**
16:    **if** $order = lower$ **then**
17:      $b \leftarrow max(F(root_u), F(root_v)$
18:    **else**
19:      $b \leftarrow min(F(root_u), F(root_v))$
20:    **end if**
21:    $d \leftarrow F(e)$
22:    $\mathcal{B}_0 \leftarrow \mathcal{B}_0 \cup \{\{(b, d)\}\}$
23:    $U.\text{LINK}(root_u, root_v)$
24: **end for**
25: **return** $(\mathcal{B}_0, E_{pos}, E_{neg}, U)$

---

## D.2   A Brief Overview of the Link-Cut Tree Data Structure

The link-cut tree data structure [55] is a well known dynamic connectivity data structure. For modifying the tree of $n$ nodes, it takes $O(\log n)$ amortized time for deleting an edge (cut) and joining two trees (link). Furthermore, it takes $O(\log n)$ amortized time for the composition of associative reductions, such as max, min, sum, on some path from any node to its root. We may view the link-cut tree data structure as a collection of trees and thus as a forest as well. Details of this forest implementation are omitted.

The link-cut tree decomposes the nodes of a tree $T$ into disjoint preferred paths. A preferred path is a sequence of nodes that strictly decreasing in depth (distance from the root of $T$) on $T$. A path has each consecutive node connected by a single edge. In particular, each node in $T$ has a single preferred child, forming a preferred edge. The maximally connected sequence of preferred edges forms a preferred path. The preferred path decomposition will change as the link-cut tree gets operated on. Each preferred path is in one to one correspondence with a splay tree [56] called an auxiliary tree on the set of nodes in the preferred path. For any node $v$ in a preferred path's auxiliary tree, its left subtree is made up of nodes higher up (closer to the root in $T$) than $v$ and its right subtree is made up of nodes lower (farther from the root in $T$) than $v$. Each auxiliary tree contains a pointer, termed the auxiliary tree's parent-pointer, from its root to the parent of the highest (closest to the root) node in the preferred path associated with the auxiliary tree.

The most important supporting operation to a link-cut tree $T$ is the EXPOSE() operation. The result of EXPOSE($v$) for $v \in T$ is the formation of a unique preferred path from the root of $T$ to $v$ with this preferred path's set of nodes forming an auxiliary tree. Furthermore, it results in $v$ to be the root of the auxiliary tree it belongs to. The complexity of EXPOSE($v$) is $O(\log n)$. For implementation details, see [55].

Let $T_1, T_2$ be two link-cut trees and $u \in T_1, v \in T_2$ with $u$ a root of $T_1$. Define the operation LINK($T_1, (u, v), T_2$) as the operation that attaches $T_1$ to $T_2$ by connecting $u$ with $v$ by an edge and outputs the resulting tree. This is achieved by simply calling EXPOSE($u$) then EXPOSE($v$), which

makes $u$ and $v$ the roots of their respective auxiliary trees. In the auxiliary tree of $u$, then set the left child of $u$ to $v$.

Let $T$ be a link-cut tree and $u, v \in T$ connected by an edge with $v$ higher up in $T$, closer to the root of $T$. Define the operation $\text{CUT}(T, (u, v))$ as the operation that disconnects $T$ by deleting the edge between $u$ and $v$. This is achieved by simply calling $\text{EXPOSE}(u)$ and then making $u$ a root by making the left child of $v$ point to $null$.

Let $T$ be a link-cut tree and $u, v \in T$. Define the operation $\text{LCA}(T, (u, v))$ as the operation that finds the least common ancestor of $u$ and $v$ in $T$. This is achieved by calling $\text{EXPOSE}(u)$ then $\text{EXPOSE}(v)$ and then taking the node pointed to by the parent-pointer of the auxiliary tree of which $u$ is root.

Let $T$ be a link-cut tree and $u, v \in T$ with $v$ higher up in the tree $T$, meaning that it is closer to the root than $u$, and there being a unique path of monotonically changing depth in the tree from $u$ to $v$. Define the operation $\text{PATH}(u, v)$ as the operation that returns a linked list of the path from $u$ to $v$ in $T$. If $v$ is the root, call $\text{EXPOSE}(u)$ and return the linked list formed by the splay tree with $u$ at root. Otherwise, first find the parent $v'$ of $v$. The parent of $v$ can be obtained by calling $\text{EXPOSE}(v)$ then traversing the splay tree it is a root of for its parent in $T$. Call $\text{EXPOSE}(u)$ to form a preferred path from u to the root of $T$ then $\text{EXPOSE}(v')$ to detach $v'$ from this preferred path. Let $\text{SPLAY}(u)$ be the operation that rotates the unique splay tree, or preferred path, containing $u$ so that $u$ becomes the root of its splay tree. After calling $\text{SPLAY}(u)$, $u$ becomes the root of a linked-list splay tree. It is a linked-list since $u$ is the lowest (farthest from the root) node in its splay tree and the rest of the preferred path is made up of a path of strictly decreasing distance to the root. Return this linked-list splay tree as the resulting path from $u$ to $v$.

Let $T$ be a link-cut tree, $u, v \in T$ with $v$ higher up in the tree than $u$ (it is closer to the root of $T$ than $u$) and there being a unique path of monotonically changing depth in the tree from $u$ to $v$. Define $\text{REDUCE}(T, u, v, op)$ to be an associative reduction on the path from $u$ to $v$. To do this, apply $\text{EXPOSE}(u)$ then $\text{EXPOSE}(v)$, then apply the associative operation on the whole auxiliary tree rooted at $u$, as implemented on a splay tree in [56]. The associative reduction takes $O(\log n)$ time. This splay tree corresponds to the preferred path from $u$ to $v$ formed from the two $\text{EXPOSE}$ operations. Notice that $\text{EXPOSE}(u)$ results in a preferred path from $u$ to the root while the second call $\text{EXPOSE}(v)$ detaches the path from $v$ to the root of $T$ from the preferred path of $u$ to the root.

Let $T$ be a link-cut tree, $u, v \in T$ and $lca$ the least common ancestor of $u, v \in T$. Assume the nodes are labeled by a pair of their value and index. Two nodes are compared by their respective values. Define $\text{ARGMAXREDUCECYCLE}(T, u, v, lca)$ as the operation that finds the edge with one of its nodes containing the maximum value on the cycle formed by $u, v$ and $lca$. There are many ways to implement this. We describe a method that maintains the $O(\log n)$ complexity of link-cut tree operations. We first compute $(value(w_1), w_1) := \text{REDUCE}(T, u, lca, max)$ to find the maximum value node along the path from $u$ to $lca$, then compute $(value(w_2), w_2) := \text{REDUCE}(T, v, lca, max)$ to find the maximum value node along the path from $v$ to $lca$. Let $w$ to be the maximum valued vertex between $w_1$ and $w_2$. If $w \neq lca(u, v)$, then find the parent $z$ of $w$; otherwise, apply $\text{EXPOSE}(u)$ then $\text{EXPOSE}(v)$ and keep track of the child $z$ of $w$ that gets detached during $\text{EXPOSE}(v)$. Parent of $w$ can be found by $\text{EXPOSE}(w)$ then traversing its splay tree to find the parent of $w \in T$. The edge $(z, w)$ is returned by $\text{ARGMAXREDUCECYCLE}(T, u, v, lca)$

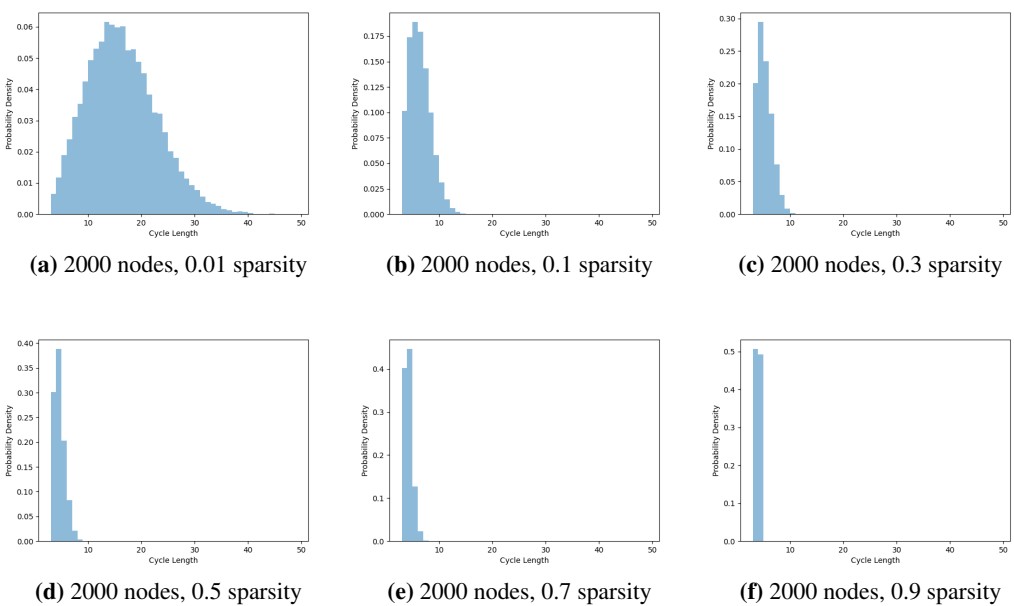

**Figure 8:** Cycle length histograms of the cycle representatives output by the extended persistence algorithm on sampled Erdos-Renyi graphs

# E Cycle Length Distribution of the Cycle Basis found by Extended Persistence Algorithm for Erdos-Renyi Graphs

We perform an experiment to determine the cycle length distribution of cycle representatives output by our algorithm on random Erdos-Renyi graphs. We observe that, as the graph becomes more dense, the distribution of cycles shifts towards very short cycles. We also find that the cycle lengths for most Erdos-Renyi sparsity hyperparameters rarely become very long. We hypothesize that the cycle basis found by the extended persistence algorithm is close to the minimal (in cycle lengths) cycle basis.

For a given node count $n$, edge count $m$, and sparsity hyperparameter, $0 \leq s \leq 1$ which we define as the Erdos-Renyi probability for keeping an edge from a clique on $n$ nodes, we sample three Erdos-Renyi graphs. We collect the multiset of $m - n + 1$ cycle lengths in the cycle basis found by the algorithm. This multiset can be visualized as a histogram. Each histogram is a relative frequency mixture of the three cycle length histograms for each graph. See Figure 8 for the histograms we obtained from sampled Erdos-Renyi graphs. Notice that, even for $0.01$ sparsity, Erdos-Renyi samples of graphs on 2000 nodes have the average cycle length of 15, which is $0.75\%$ of $n = 2000$.

To put this in perspective, assume that we can relate the Erdos-Renyi sparsity $s$ by $\hat{s} := \frac{m}{n^2}$. For the datasets of our experiments, we have $\hat{s} \approx 0.009, 0.0048, 0.39, 0.062, 0.084, 0.0018, 0.032,$ and $0.045$ for DD, PROTEINS, IMDB-MULTI, MUTAG, PINWHEELS, 2CYCLES, MOLBACE, and MOLBBBP, respectively. The sparsity estimator is in the range of $0.0018 \leq \hat{s} \leq 0.39$, which tells us that most of the cycle lengths found by our algorithm are short.

## F Rational Hat Function Visualization

Figure 9 and Figure 10 visualize the rational hat function for fixed $r$ value and varying $x$ and $y$ values. Notice the boundedness of the plot as $(x, y) \to \infty$. For the theory behind the rational hat function, see [1].

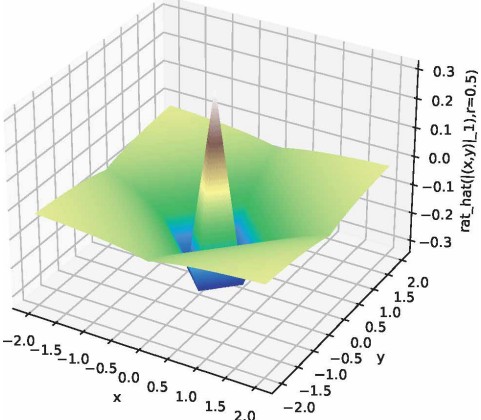

**Figure 9:** The function $\hat{r}$, output sliced at one dimension, as a function of $|(x, y)|_1$ with $r = 0.5$ from Equation 1. The point $(x, y)$ is given by $(x, y) = \mathbf{p} - \mathbf{c}$.

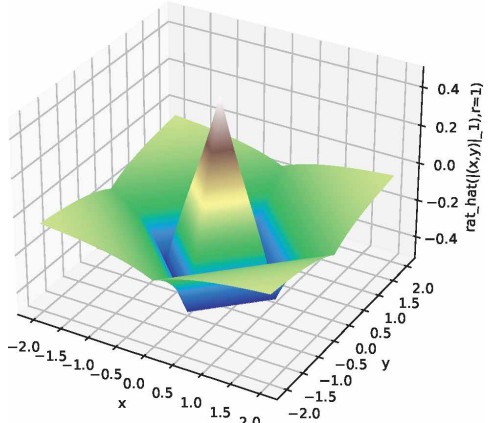

**Figure 10:** The function $\hat{r}$, output sliced at one dimension, as a function of $|(x, y)|_1$ with $r = 1.0$ from Equation 1. The point $(x, y)$ is given by $(x, y) = \mathbf{p} - \mathbf{c}$.

## G Datasets and Hyperparameter Information

Here are the datasets, both synthetic and real world, used in all of our experiments along with training hyperparameter information.

The barcode vectorization layer, or concatenation of four-rational hat functions, is set to a dimension of 256. The LSTM used on the explicit cycle representatives was set to a 2-layer bidirectional LSTM with single channel inputs and 256 dimensional vector representations. Due to the fact that our algorithm on random Erdos-Renyi graphs rarely encounters long cycles, we set the LSTM layers to a small number like 2 to avoid overfitting.

| Dataset and Hyperparameter Information | | | | | | | | |
|---|---|---|---|---|---|---|---|---|
| Dataset | Graphs | Classes | Avg. Vertices | Avg. Edges | lr | Node Attrs.(Y/N) | num. layers | Class ratio |
| DD | 1178 | 2 | 284.32 | 715.66 | 0.01 | Yes | 2 | 691/487 |
| PROTEINS | 1113 | 2 | 39.06 | 72.82 | 0.01 | Yes | 2 | 663/422 |
| IMDB-MULTI | 1500 | 3 | 13.00 | 65.94 | 0.01 | No | 2 | 500/500/500 |
| MUTAG | 188 | 2 | 17.93 | 19.79 | 0.01 | Yes | 1 | 63/125 |
| PINWHEELS | 100 | 2 | 71.934 | 437.604 | 0.01 | No | 2 | 50/50 |
| 2CYCLES | 400 | 2 | 551.26 | 551.26 | 0.01 | No | 2 | 200/200 |
| MOLBACE | 1513 | 2 | 34.09 | 36.9 | 0.001 | Yes | 2 | 822/691 |
| MOLBBBP | 2039 | 2 | 24.06 | 25.95 | 0.001 | Yes | 2 | 479/1560 |

**Table 3:** Dataset statistics and training hyperparameters used for all datasets in scoring experiments of Table 1 and Table 2

## H Implementation Dependencies

Our experiments have the following dependencies: python 3.9.1, torch 1.10.1, torch_geometric 2.0.5, torch_scatter 2.0.9, torch_sparse 0.6.13, scipy 1.6.3, numpy 1.21.2, CUDA 11.2, GCC 7.5.0.

# I  Visualization of Graph Filtrations

We visualize the filtration functions $f_G$ learned on graphs $G$ for the datasets: IMDB-MULTI, MUTAG, and REDDIT-BINARY. The value of $f_G(v)$ for each $v \in V$ is shown in each figure.

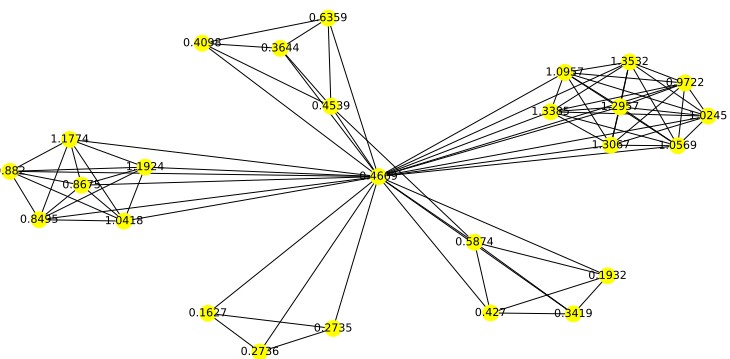

**Figure 11:** IMDB-MULTI learned filtration function

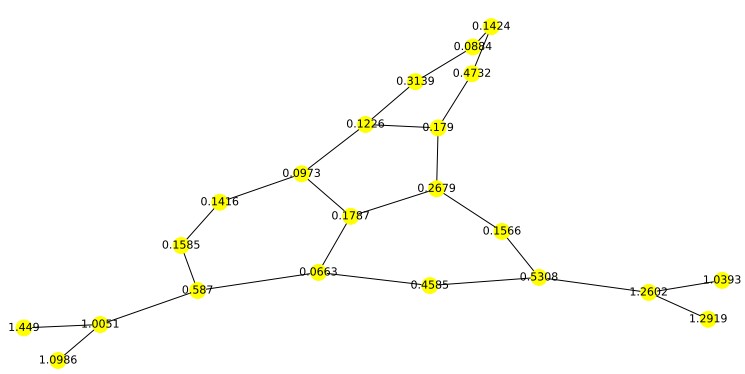

**Figure 12:** MUTAG learned filtration function

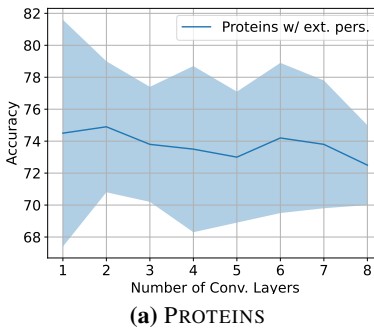
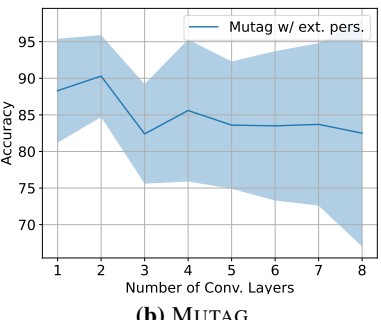

**(a)** PROTEINS

**(b)** MUTAG

**Figure 13:** An exhibit of oversmoothing in the filtration convolutional layers. Plot of the average accuracy with std. dev. as a function of the number of convolutional layers before the Jumping Knowledge MLP and the extended persistence readout. The PROTEINS and MUTAG datasets were used in (a) and (b) respectively.

## J   Additional Experiments

### J.1   Number of Convolutional Layers Experiment

We also perform an experiment to determine the number of layers in the MPGNN of the filtration function that has the highest performance. Due to oversmoothing [57], which is exacerbated by the required scalar-dimensional vertex embeddings, as we increase the number of layers for the filtration function the performance drops. See Figure 13 for an illustration of this phenomenon on the PROTEINS and MUTAG dataset. For these two datasets, two layers perform the best.

| Model | F1 Macro |
|---|---|
| GEFL-bars | 84.0±14.2 |
| GEFL-bars+cycles | **85.4±10.3** |
| GCN | 69.8 ±15.2 |
| GraphSAGE | 69.5±11.0 |
| GIN0 | 79.9±9.0 |
| GIN | 84.9±6.0 |

**Table 4:** MUTAG Macro F1 Scores

### J.2   Macro F1 Experiments

We perform further experiments with the MUTAG dataset and evaluate performance with the F1 Macro score due to the class imbalance in the MUTAG dataset, see Table 3. See Table 4 for details of the F1 Macro evaluation.

### J.3   Experiments on Transferability

We perform an experiment on the transferability of our approach from real world datasets to their edge corrupted versions and compare with the standard GNN baselines. We form a corrupted version of the data by applying a Bernoulli random variable with $p = 0.2$ on each edge of the MUTAG and IMDB-MULTI datasets. We perform zero-shot, one-shot and 10-shot transfer learning. We include an ablation study to compare with models without fine-tuning and with $0, 1, 10$ and $100$ epochs of training.

We notice that our model does not directly transfer in either zero or one shot transfer learning. However after 10 epochs of fine-tuning, there appears to be an advantage to transfer learning over raw training. For the MUTAG dataset, there is a $77.7 - 64.2$ average Macro F1 score difference for a bars only representation model and a $74.8 - 69.8$ average Macro F1 score difference for bars with cycle representatives. Furthermore, on MUTAG the baseline GNNs do not show any improvement with 10-shot transfer learning. For IMDB-MULTI there is a $48.3 - 46.1$ average accuracy difference for

bars only representation model and a $49.7 - 48.0$ average accuracy difference for the bars with cycle representatives model. As for MUTAG, for IMDB-MULTI there is no transfer learning improvement for the baselines.

We hypothesize that both versions of our model transfer well since our model captures global topological information with low variance. This is more conducive to transfer learning than an aggregation of each node's local neighborhood information as in the baselines. The baselines result in training a higher variance classifier. Low variance classifiers transfer easier since their decision boundary is easier to adapt to new data. We also hypothesize that the bars only model transfers better than the bars with cycle representatives model due to its even lower variance, no LSTM parameters.

Our model downsamples the graph representation into scalars (a set of sequences of scalars and a set of pairs of scalars) upon computing bars and cycle representatives. Due to our architecture, to accommodate these scalars, this results in a higher bias classifier with few parameters. Due to the bias variance trade off, the variance should be low.

| Transfer learning results | | | | | | | | |
|---|---|---|---|---|---|---|---|---|
| Dataset | Pretraining (Epochs) | Finetuning (Epochs) | Ours+bars | Ours+bars+cycles | GCN | GraphSage | GIN0 | GIN |
| MUTAG-0.2 (F1 Macro) | 100 | – | 83.2±9.6 | 78.3±17.3 | 73.4±10.4 | 74.5±09.1 | 85.6±05.7 | 83.5±11.2 |
| MUTAG-0.2 (F1 Macro) | 100 | 0 | 31.4±11.2 | 29.4±11.4 | 32.4±7.9 | 39.9±0.9 | 35.4±7.3 | 35.4±7.3 |
| MUTAG-0.2 (F1 Macro) | 0 | – | $37.1 \pm 12.2$ | $32.7 \pm 7.9$ | $34.6 \pm 8.7$ | $36.1 \pm 7.8$ | $35.8 \pm 10.1$ | $34.3 \pm 7.1$ |
| MUTAG-0.2 (F1 Macro) | 100 | 1 | 39.9±0.9 | 39.9±0.9 | 39.9±0.9 | 39.9±0.9 | 39.9±0.9 | 39.9±0.9 |
| MUTAG-0.2 (F1 Macro) | 1 | – | $39.9 \pm 0.9$ | $39.9 \pm 0.9$ | $39.9 \pm 0.9$ | $39.9 \pm 0.9$ | $39.9 \pm 0.9$ | $39.9 \pm 0.9$ |
| MUTAG-0.2 (F1 Macro) | 100 | 10 | 76.7±9.7 | 74.8±10.2 | 39.9±0.9 | 39.7±14.3 | 39.9±0.9 | 39.9±0.9 |
| MUTAG-0.2 (F1 Macro) | 10 | – | $64.2 \pm 17.9$ | $69.8 \pm 17.7$ | $43.9 \pm 9.5$ | $44.2 \pm 9.7$ | $41.8 \pm 6.2$ | $44.1 \pm 13.3$ |
| IMDB-MULTI-0.2 (Acc.) | 100 | – | 49.2±3.9 | 50.3±2.9 | 49.4±2.5 | 50.5±2.6 | 49.6±3.1 | 51.5±3.1 |
| IMDB-MULTI-0.2 (Acc.) | 100 | 0 | 33.1±1.1 | 32.5±3.1 | 16.8±0.5 | 20.2±4.9 | 24.0±6.1 | 21.0±7.9 |
| IMDB-MULTI-0.2 (Acc.) | 0 | – | $34.5 \pm 4.7$ | $34.3 \pm 2.4$ | $20.2 \pm 4.3$ | $19.6 \pm 4.3$ | $20.4 \pm 4.3$ | $20.6 \pm 5.0$ |
| IMDB-MULTI-0.2 (Acc.) | 100 | 1 | 39.7±5.3 | 40.7±5.9 | 44.8±4.4 | 50.5±3.5 | 34.3±4.2 | 40.9±3.4 |
| IMDB-MULTI-0.2 (Acc.) | 1 | – | $34.3 \pm 1.6$ | $38.8 \pm 4.9$ | $47.2 \pm 3.5$ | $47.8 \pm 3.3$ | $43.2 \pm 3.3$ | $45.8 \pm 5.5$ |
| IMDB-MULTI-0.2 (Acc.) | 100 | 10 | 48.3±2.2 | 49.7±3.0 | 48.9±3.3 | 50.5±3.5 | 46.7±2.9 | 47.9±4.5 |
| IMDB-MULTI-0.2 (Acc.) | 10 | – | $46.1 \pm 2.1$ | $48.0 \pm 3.1$ | $50.1 \pm 3.1$ | $49.9 \pm 1.9$ | $51.5 \pm 2.5$ | $50.1 \pm 2.5$ |

**Table 5:** We list here the scores for our transfer learning experiments on MUTAG and IMDB-MULTI. We pretrain on the original MUTAG and IMDB-MULTI by 10-fold cross validation. Then, we fine-tune the pretrained models on corrupted versions of these datasets also by 10-fold cross validation. These two corrupted datasets are obtained by filtering by a Bernoulli random variable of $p = 0.2$ on each edge. This is very likely to introduce different cycle patterns in the data. Fine-tuning is performed for $0$, $1$ and $10$ epochs, while '–' denotes no finetuning.

