# OpenReview forum: "GEFL: Extended Filtration Learning for Graph Classification"
_logconference.io/LOG/2022/Conference — LoG 2022 Poster_

### Official Review · Reviewer_yrKA · 2022-10-08

**Overall Score:** 6
**Confidence:** 4

**Review:**

**Summarization**

The paper introduces a graph classification framework based on the theory of extended persistent homology.  The proposed framework incorporates the extended persistence diagrams (EPDs) as well as the cycle representation to enhance the model's learning power. In addition, the paper introduces the link-cut tree structure to a previous algorithm to accelerate the computation of EPDs.


Generally speaking, the proposed algorithm and framework are new, and the experiment is acceptable. Although the improvement is not so surprising (as it directly builds upon several previous works), I still appreciate the contribution of the paper.



**Pros**

The proposed algorithm is new. And the efficiency is well evaluated by comparison with the popular TDA tool gudhi.

The framework that incorporates both the cycle representation and EPDs is new. And the theoretical analysis on its expressiveness is also provided.

The experiment is acceptable as it incorporates various popular baselines and benchmarks. Also, the effectiveness of the proposed framework is justified on these benchmarks.

**Cons**

Certain improvements are not so surprising. And the synthetic datasets seem just like toy examples.

The paper improves a previous work [3] by introducing a tree-like data structure to the proposed algorithm. It is new, but the improvement is directly built upon a previous algorithm, and thus somewhat not so interesting.

The framework incorporates both the EPDs and the cycle representation to enhance the learning power. But it seems like a direct combination of two previous works (EPD encoding from [4], and cycle representation from [24]).

The synthetic datasets are like just toy examples, as the involved cycle representation generally contains the information of cycles, which is the key to these datasets. Therefore, the framework with cycle representation is guaranteed to perform better on these datasets. Is it possible to analyze some other properties of the model, e.g., transferability? Or, EPDs are known to capture the "shallowest cycles" in terms of filtration (Yan et al.), i.e., among all cycles having $e_1$ as the highest edge, $e_2$ is the lowest edge in a cycle such that this lowest value is the highest possible. Is it possible to design a dataset that aims to predict the information of non-shallowest cycles using EPDs (since it is not the information directly encoded in EPDs, or the cycle representation from EPD encoding)?

Some references on accelerating persistent homology computation should also be added, e.g., (de et al., Yan et al.).

(de et al.) de Surrel T, Hensel F, Carrière M, et al. RipsNet: a general architecture for fast and robust estimation of the persistent homology of point clouds. GTRL 2022.

(Yan et al.) Yan Z, Ma T, Gao L, et al. Neural Approximation of Graph Topological Features. Neurips 2022.

**Questions**

Actually, I'm surprised to find that adding cycle representation helps the model perform better, as EPDs should contain the information of cycles (or say, a cycle basis). From my perspective, the improvement comes from the fact that EPDs only contain part of the cycle information (the largest and lowest persistence value, or, the birth and death time, and that is why certain information such as the length of cycles are missing), while the cycle representation contains all the information. Could you please explain the reason for the improvement?

---

### Official Review · Reviewer_76Cm · 2022-10-20

**Overall Score:** 8
**Confidence:** 4

**Review:**

Summary:

The paper proposes a supervised learning model for graph classification based on extended persistent homology. A learnable filtration function drives the computation of extended persistence resulting to persistence diagrams and cycle representatives, vectorizations of which act as a global graph readout layer for graph classification. Theoretical results prove the enhanced expressivity of extended persistence compared to standard persistent homology, along with the ability of the proposed extended persistence-based model to infer distinguishing topological statistics of graphs, namely, number of connected components and cycle sizes. A parallel algorithm is also presented for the computation of extended persistence, which is improving upon prior work. Experimental results verify the distinguishing abilities of the proposed model against the WL[1] test.

Comments:

The novelty of the proposed model, the clarity of exposition, and the exhaustive experimental and theoretical evaluation contribute to my vote for accepting the paper. Specifically:
- The use of extended persistence for graph classification is novel and well justified.
- The motivation and contributions are clearly stated.
- Relevant literature has been exhaustively identified, both during the method presentation and the experimental evaluation, in terms of baselines.
- Figures are clear and informative, aiding to the understanding of extended persistence, as well as the model pipeline.
- The proposed model is thoroughly and clearly presented, along with justification of architectural choices.
- The pseudocode of the extended persistence computation algorithm is nicely and clearly explained.
- The proposed model is thoroughly evaluated against a plethora of baselines and datasets, with exceptional results.

A few suggestions that would improve the quality of exposition and evaluation:
1. In Section 4.1 it is explicitly stated that the algorithm improves upon the work of Yan et al. [3]. I was expecting to see timing comparisons between the proposed algorithm and [3], with and without cycle representative computations, so that the effects of paralellization and data structure choice would shine.
2. It would be nice to include an example of cycle representatives, as computed by the proposed algorithm, in Figure 1.
3. A more elaborate justification for the choice of LSTM for encoding the (variable size) cycle representations would be beneficial.
4. Since the proposed algorithm heavily depends on the union-find and link-cut tree data structures, a short description of both in the appendix would aid exposition.

Typos:
- Line 190, "componenents".
- Line 238, $\mathcal{B}_0^{upper}$ instead of $\mathcal{B}_0^{up}$.
- Line 293, missing Table reference.

---

### Official Review · Reviewer_PA8Q · 2022-10-22

**Overall Score:** 8
**Confidence:** 4

**Review:**

This paper introduces extended persistence into the supervised learning framework, and brings in global connected component and cycle measurement information into the graph representations. It is also efficient and scales. The performance on the real world data and synthetic data shows the superiority of the method. The work is innovative and solid. But the experiment has weakness: average accuracy is used as the measure, but it cannot reflect the real performance in imbalanced classification. F1 for each class or macro F1 are better measures.

---

### Official Review · Reviewer_4SLT · 2022-10-22

**Overall Score:** 5
**Confidence:** 4

**Review:**

Summary of contribution:
This paper tackles the problem of graph classification using extended
persistence, which enables the homology generators to always have finite height.
In doing so, an improvment in the computation of extended persistence is
proposed.

Strengths:
* The practical improvements on computing extended persistence are great!

Weaknesses:
* The paper needs some reoganization.  For example:
    - Section 2 assumes that the reader knows what is presented in Section 3,
        and many terms are used before they are defined.
    - The filtration define in lines 60-62 is an index filtration.  In general,
        it might be helpful to talk about the filter (sequence of simplices) that is
        compatible with a function on the vertices, and make the definitions
        specific to how they are used rather than going back and forth between
        general definitions and setting-specific definitions.
    - Relative persistence is never really defined.  How are the two filtrations
        "concatenated?"
    - Often, variables are defined after they are used.  For equations, this is
        ok for "obvious" variables, but this paper seems to make it the norm to
        always define variables after they are used, which is confusing to the
        reader.
* The paper can be more precise in parts.  For example:
    - on line 30 the "high computational cost" of extended persistence is
        mentioned, but not explicitly stated what that cost is.
    - Line 100 talks about a "vertex embedding" but does not specify where the
        vertices are embedded.
    - Line 106 (and elsewhere): permutation invariant is not explicitly defined.
        permutations of what?
    - Line 164: what does it mean to sort the filtration stably?  (and what does
        it mean to sort a filtration?)
    - In Algorithm 1, Lines 8 and 12, the order matters, however, the pseudocode
        allows the edges to be processed in any arbitrary order.
* Line 97-98: the symmetry within the diagram is known.  It is not due to the
    facut that there are O(n) filtration values and O(m) bars.

Some relevant references are missing:
* Keros, Nanda, and Subr's "Dist2Cycle: A Simplicial Neural Network for Homology Localization"
* Levy et al.'s "Topological Feature Extraction and Visualization of Whole Slide Images using Graph Neural Networks"
* Montufar, Otter, and Wang's "Can neural networks learn persistent homology features?"
* Yan et al. "Neural Approximation of Graph Topological Features"
* Zhao, Ye, Chen, and Wang's "Persistence Enhanced Graph Neural Network"

Recommendation:
My recommendation is weak accept.  While extended persistence is introduced in
this setting, this is a natural use of extended persistence and not a novel
contribution.  However, in practice, the speedup obtained is impressive.  I have
concerns about the correctness of the algorithm, but think that it can be
addressed.

Questions to Authors:
See weaknesses above.

Additional Feedback:

* "Order" in the title should be capitalized.
* Some sentences in the paper, are ambiguous.  E.g., the 60x improvment over
    state-of-the art in the abstract is unclear if this is in reference to the
    computational of extended persistence or to the problem of graph classification.
* Throughout, variables are intruduced in non-standard ways.  For example, on
    line 57, it should either first define n,m, then define G to be a graph with
    n vertices and m edges OR it should define G then let n=|V| and m=|E|.  The
    way it is written, the logic does not flow correctly.
* lines 73-76: using both (b_i,d_i) and [b_i,d_i] for a bar is a bit confusing,
    especially becuase the bar is actually the half-open interval [b_i,d_i).
* Talking about "the four barcodes" is a bit deceiving, as there is really only
    one barcode, but it's partitioned into four.  I recommend using the term
    "sub-barcodes" for this reason.
* "well-defined" should be hypenated.
* line numbers are off-by one when referencing lines of the algorithm.
* In Alg. 1, the input/output should not be labeled as lines of the algorithm.
* With the comments of lines 279-284, I wonder if using
    Augmented/verbose persistence diagrams could be helpful here.  (See the ArXiv
    paper "A Faithful Discretization of the Augmented Persistent Homology Transform"
    for a definition of Augmented Persistnece Diagram or "Ephemeral persistence
    features and the stability of filtered chain complexes" for a definition of
    verbose diagrams; basically, the "on-diagonal" points that are computed are not
    omitted from the representation of the persistence diagram).
* Line 293: table reference is broken.

Type of paper: Full paper proceedings track submission (max 9 main pages). This
requirement is met.

---

### Meta-Review · Area_Chair_5vyi · 2022-11-16

**Confidence:** 5
**Recommendation:** Accept

**Meta Review:**

This paper studies graph classification from the perspective of extended persistent homology, providing both practical and (minor) theoretical advances to make tools for this task more usable.  Almost all reviewers agree on the value and interest of the paper as well as its fit to this conference.  The authors were extremely responsive during the rebuttal phase and addressed many expository concerns and missing numbers highlighted by the reviewers, further improving the work.

Positive
* Practical improvements on computing extended persistence
* Extended persistence not novel, although this is a new application and practical contribution

Negative
* Expository issues, revised in version uploaded during discussion phase

Additional comments
* Please include the tables of values sent in response to PA8Q/76Cm/yrKA in the final version of the paper, e.g. as an appendix or supplemental document

---

### Decision · Program_Chairs · 2022-11-22

Accept (Poster)